# A Message Passing-Assisted Iterative Noise Cancellation Method for Clipped OTFS-BFDM Systems

**DOI:** 10.3390/s22103937

**Published:** 2022-05-23

**Authors:** Tingyao Wu, Hongxia Bie, Jinfang Wen

**Affiliations:** 1School of Artificial Intelligence, Beijing University of Posts and Telecommunications, Beijing 100876, China; wutingyao@bupt.edu.cn; 2School of Electronic Information, Wuhan University, Wuhan 430072, China; wenjinfang@whu.edu.cn

**Keywords:** OTFS-BFDM, high-mobility scenarios, message passing, OCF, PARR

## Abstract

Compared with orthogonal frequency division multiplexing (OFDM) systems, orthogonal time frequency space systems based on bi-orthogonal frequency division multiplexing (OTFS-BFDM) have lower out-of-band emission (OOBE) and better robustness to high-mobility scenarios, but suffer from a higher peak-to-average ratio (PAPR) in large data packets. In this paper, one-iteration clipping and filtering (OCF) is adopted to reduce the PAPR of OTFS-BFDM signals. However, the extra noise introduced by the clipping process, i.e., clipping noise, will distort the desired signal and increase the bit error rate (BER). We propose a message passing (MP)-assisted iterative cancellation (MP-AIC) method to cancel the clipping noise based on the traditional MP decoding at the receiver, which incorporates with the (OCF) at the transmitter to keep the sparsity of the effective channel matrix. The main idea of MP-AIC is to extract the residual signal fed to the MP detector by iteratively constructing reference clipping noise at the receiver. During each iteration, the variance of residual signal and channel noise are taken as input parameters of MP decoding to improve the BER. Moreover, the convergence probability of the modulation alphabet after MP decoding in the current iteration is used as the initial probability of MP decoding in the next iteration to accelerate the convergence rate of MP decoding. Simulation results show that the proposed MP-AIC method significantly improves MP-decoding accuracy while accelerating the BER convergence in the clipped OTFS-BFDM system. In the clipped OTFS-BFDM system with rectangular pulse shaping, the BER of MP-AIC with two iterations can be reduced by 72% more than that without clipping noise cancellation.

## 1. Introduction

Fifth-generation (5G) mobile communications are envisioned to accommodate many emerging applications in high-mobility scenarios, such as high-speed train, vehicle-to-vehicle, and vehicle-to-infrastructure communications [1]. In the future, sixth-generation (6G) mobile communications will enter the terahertz band, enabling a fully connected, intelligent digital world [2,3]. In all of the above, the wireless channel manifests strong Doppler shifts that increase with the carrier frequency or terminal speed. As a result, conventional orthogonal frequency division multiplexing (OFDM) modulation cannot provide reliable transmission due to its sensitivity to high-Doppler shifts [4]. Recently, orthogonal time frequency space (OTFS) modulation systems have been introduced, which are more robust to Doppler spread compared to OFDM-based systems [5]. Since the channel and signal multiplexing of OTFS systems are represented in the delay Doppler (DD) domain rather than the traditional time-frequency (TF) domain, the fast time-varying channel appears almost non-fading as observed from the DD domain [6]. In other words, each symbol in a frame experiences the same channel again, significantly reducing the overhead and complexity associated with physical layer adaptation. Another attractive feature of OTFS systems is that the equivalent channel matrix (ECM) is sparse, enabling one to use low-complexity detection algorithms at the receiver [7].

Although the OTFS system has the above advantages, its inherent rectangular prototype window function (PWF) has high out-of-band emission (OOBE), resulting in low spectrum utilization in multi-user scenarios [8]. In [9], the OTFS system employs a long ellipsoid function as the PWF, achieving a significant OOBE reduction. However, the channel gains of edge symbols become lower, which increases the bit error rate (BER). In [10], an extended general frequency division multiplexing (GFDM) framework was proposed to replace OTFS for low OOBE applications in high-Doppler multi-user scenarios. However, when GFDM uses zero-forcing detection for perfect reconstruction (PR), the BER obtained will be worse due to the sensitivity of inverse matrix to noise interference [11]. As a variant of GFDM, bi-orthogonal frequency division multiplexing (BFDM) modulation was proposed in [12,13], which demodulates the signal by using the matched filter coefficient matrix to avoid the noise interference on the inverse matrix. A new idea is to use BFDM as the time-frequency modulation in the OTFS system [14]. The BFDM-based OTFS (OTFS-BFDM) can provide low OOBE without any loss in BER when using the Dirichlet PWF.

Like the OTFS signal, the OTFS-BFDM signal also has an abnormally high peak-to-average ratio (PAPR) in big data packets. When the linear dynamic range of the high power amplifier (HPA) is insufficient, a large PAPR will cause severe distortion of the transmission signal [15]. Currently, various techniques have been proposed to reduce the PAPR, including clipping and filtering (CF) [16,17,18], compression-expansion transformation [19,20], and selective mapping (SLM) [21,22]. It is worth noting that for PAPR reduction in multi-carrier systems using message-passing (MP) decoding, the sparsity of the ECM must be maintained [23]. In [24], Rajasekaran et al. proposed an SLM method based on a resource allocation strategy to suppress the PAPR of sparse code multiple access-based OFDM (SCMA-OFDM), but extra sideband information must be transmitted. In [25], Gao and Zheng adjusted the clipping filter coefficient to achieve a good tradeoff between the PAPR and BER performance of the pilot-embedded OTFS system. However, the coefficient is filled in the guard region to avoid interferences between data and pilot signals, degrading the spectral efficiency. In [26], an MP-assisted (MP-A) clipping method was developed to reduce the PAPR of SCMA-OFDM system. However, the clipping is limited to the Nyquist sampling rate, and the clipping noise is slightly mitigated rather than canceled.

Motivated by these, at the receiver, we propose an MP-assisted iterative cancellation (MP-AIC) method to cancel the clipping noise in clipped OTFS-BFDM systems. Furthermore, at the transmitter, one-iteration clipping and filtering (OCF) [26] is used to prevent peak regeneration and maintain the sparsity of ECM. In the MP-AIC, the reference clipping noise is first constructed, then the revised received signal is obtained by subtracting the reference clipping noise from the initial received signal, and finally the residual signal is extracted from the revised received signal for feedback to the MP detector. Through several iterations, the influence of clipping noise on MP is minimized, and the clipping noise can be considered to be eliminated. Consequently, this method can further improve the decoding accuracy compared with the MP-A method and accelerate the BER convergence without compromising the spectral efficiency.

The key contributions of this paper can be summarized as follows:The OTFS-BFDM input–output relationship is formulated in matrix form, and the problem of high PAPR in OTFS-BFDM systems is presented by simulation.The OCF method for PAPR reduction is considered to maintain the sparsity of ECM and the statistical properties of clipping noise are analyzed.The proposed MP-AIC method is described in detail, where the residual signal is calculated, and initial probability of MP decoding in each iteration is the convergence probability of the next iteration.

The remainder of the paper is organized as follows. Section 2 introduces the OTFS-BFDM system and the problem of PAPR. Section 3 elaborates the proposed MP-AIC method of Clipping Noise. The simulation results and analysis is presented in Section 4 and the paper is concluded in Section 5. In addition, we use mathematical symbols as shown in Table 1 throughout the paper.

## 2. OTFS-BFDM System

### 2.1. OTFS-BFDM System Model

We assume that an OTFS-BFDM transmission frame occupies *M* sub-carriers and *N* time slots. The sub-carrier spacing is Δf and the duration of a time slot is *T*, where Δf=1/T. When D[m,n] is the n-th sub-symbol on the m-th sub-carrier in the TF domain, the corresponding DD domain signal d[l,k] (e.g., 4QAM symbol) is quasi-period with delay period τ=1/Δf and Doppler period v=1/T. Figure 1 indicates the OTFS-BFDM system model.

#### 2.1.1. Transmitter

On the DD plane, all symbols on each transmission frame can be composed as an MN×1 vector represented by:(1)$$d={\left[d_{0}^{T},d_{1}^{T},\cdots \;,d_{N-1}^{T}\right]}^{T},$$
where $d_{i}={\left[d[0,i],d[1,i],\cdots \;,d[M-1,i]\right]}^{T}$, 0≤i≤N−1.

The QAM symbol vector d in the DD domain can be mapped to D in the TF domain by the inverse symplectic finite Fourier transform (ISFFT) transform, as follows:(2)$$D=(F_{N}^{-1}⊗F_{M})d.$$

Since the BFDM modulation follows the cyclic convolutional filter bank (CCFB) structure, the BFDM modulation matrix can be obtained as:(3)$$A=\underset{P}{\underbrace{circ\{{\left[P_{0},P_{N-1},\cdots \;,P_{1}\right]}_{M\times 1}\}}}\times diag\left\{{\left[F_{M}^{-1},F_{M}^{-1},\cdots \;,F_{M}^{-1}\right]}_{N\times 1}\right\}$$
by filtering each sub-carrier through the polyphase network.

In (Equation 3), P is an L×L block-circulant analysis filter coefficient matrix of the BFDM modulation, where p(ℓ) is the PWF of the analysis filter bank with length L=MN. The matrix $P_{i}$ consists of p(ℓ) and can be expressed as:(4)$$P_{i}=diag\left\{p\left(iM\right),p\left(iM+1\right),\cdots \;,p\left(iM+M-1\right)\right\},$$
where 0≤i≤N−1. In addition, according to the properties of the block-circulant matrix, P can be rewritten as:(5)$$P=\left(F_{N}^{-1}⊗I_{M}\right)\Lambda \left(F_{N}⊗I_{M}\right),$$
where $\Lambda =diag(\Lambda_{0},\Lambda_{1},\cdots \;,\Lambda_{N-1})$, $\Lambda_{n}\in C^{M\times M}$ can be calculated by:(6)$$\left\{\begin{array}{c} \Phi \left(i,j\right)={\left[P_{0}\left(i,j\right),P_{1}\left(i,j\right),\cdots \;,P_{N-1}\left(i,j\right)\right]}^{T}\\ \Lambda_{0}\left(i,j\right),\Lambda_{1}\left(i,j\right),\cdots \;,\Lambda_{N-1}\left(i,j\right)]=\sqrt{N}F_{N}\Phi \left(i,j\right)\\ 0\leq i,j\leq M-1. \end{array}\right.$$

Referring to (Equation 4) and (Equation 6), since $P_{n}\left(i,j\right)$ is a diagonal matrix and acts as a vector element of Φ(i,j), Φ(i,j)≠0 only when i=j. Moreover, the *N*-DFT transform of Φ(i,j) yields $\Lambda_{n}\left(i,j\right)$. Therefore, the elements of Λ in the main diagonal are not zero, i.e., Λ is a diagonal matrix.

The time domain signal s on the physical channel is obtained by performing BFDM modulation on D as follows:(7)$$\begin{array}{cc} s & =AD=P\left(I_{N}⊗F_{M}^{-1}\right)\left(F_{N}^{-1}⊗F_{M}\right)d\\ \; & =P(F_{N}^{-1}⊗I_{M})d. \end{array}$$

#### 2.1.2. Receiver

Since BFDM satisfies the PR condition $AB=I_{L}$, once the modulation matrix A is determined, the demodulation matrix B can be derived as follows:(8)$$B=diag\left\{{\left[F_{M},F_{M},\cdots ,F_{M}\right]}_{N\times 1}\right\}\times \underset{Q}{\underbrace{circ\{{\left[Q_{0},Q_{1},\cdots ,Q_{N-1}\right]}_{M\times 1}\}}},$$
where 0≤i≤N−1 and $Q_{i}=diag\left\{q\left(iM\right),q\left(iM+1\right),\cdots \;,q\left(iM+M-1\right)\right\}$. The PWF q(ℓ) of synthesis filter bank can be extracted from the first-row matrix element of Q. Similar to (Equation 5), the synthesis filter coefficient matrix Q can be reinterpreted as:(9)$$Q=\left(F_{N}^{-1}⊗I_{M}\right)\Lambda^{-1}\left(F_{N}⊗I_{M}\right),$$

Assume that the OTFS-BFDM signal undergoes a linear time-varying (LTV) channel with *P* discrete propagation paths. Let $h_{p}$, $l_{p}$ and $k_{p}$ denote the complex path gain, delay tap, and Doppler tap associated with the *p*-th path, respectively. Considering the sparsity of channel representation, it is convenient to express the response h(τ,v) as [27]:(10)$$h\left(\tau ,v\right)=\sum_{p=1}^{P}h_{p}\delta \left(t-\frac{l_{p}}{M\Delta f}\right)\delta \left(v-\frac{k_{p}}{NT}\right),$$
where δ(·) is the Dirac delta function.

Adding a cyclic prefix (CP) of sufficient length to s before transmission and discarding the CP at the receiver, the received time-domain signal can be presented by:(11)$$\tilde{s}=Hs+w,$$
where H is the L×L matrix
(12)$$H=\sum_{p=1}^{P}h_{p}\Pi^{l_{p}}\Delta^{k_{p}}.$$

In (Equation 12), $\Pi =circ\{{\left[0,1,0,\cdots \;,0\right]}_{L\times 1}\}$ and $\Delta =diag\{1,e^{j2\pi /L},\cdots \;,e^{j2\pi (L-1)/L}\}$. The Gaussian noise $w\in C^{L\times 1}$ is independently and identically distributed (i.i.d) with $w_{i}∼CN\left(0,\sigma_{n}^{2}\right)$, i=0,⋯,L−1.

On the receiver, the signal $\tilde{s}$ can be mapped to $\tilde{d}$ in DD domain by BFDM demodulation and symplectic finite Fourier (SFFT) transform, which is expressed as:(13)$$\begin{array}{cc} \tilde{d}= & \left(F_{N}⊗F_{M}^{-1}\right)B\tilde{s}\\ = & \underset{H_{eq}}{\underbrace{\Lambda^{-1}\left(F_{N}⊗I_{M}\right)H\left(F_{N}^{-1}⊗I_{M}\right)\Lambda }}d+\underset{\tilde{w}}{\underbrace{\Lambda^{-1}\left(F_{N}⊗I_{M}\right)w}}, \end{array}$$
where $H_{eq}\in C^{L\times L}$ is the ECM of the OTFS-BFDM system, $F_{N}⊗F_{M}^{-1}$ is the symplectic finite Fourier transform (SFFT) transform, and $\tilde{w}\in C^{L\times 1}$ represents the channel noise vector.

When OTFS-BFDM adopts the shorter rectangular PWF, Λ evolves into an L×L identity matrix. The OTFS-BFDM input-output relationship is re-expressed as:(14)$$\begin{array}{cc} \tilde{d} & =\left(F_{N}⊗F_{M}^{-1}\right)B\tilde{s}\\ \; & =\underset{H_{eq}^{Rec}}{\underbrace{\left(F_{N}⊗I_{M}\right)H\left(F_{N}^{-1}⊗I_{M}\right)}}d+\left(F_{N}⊗I_{M}\right)w. \end{array}$$

In (Equation 14), $H_{eq}^{Rec}$ is the ECM of the OTFS system, given in [7] to be sparsely connected. Because $\Lambda^{-1}$ is a diagonal matrix, $H_{eq}$ in (Equation 13) is also sparsely connected.

### 2.2. CCDF of PAPR

The OTFS-BFDM system extends the waveform flexibility by using long non-rectangular PWFs such as root raised cosine (RRC), raised cosine (RC), and Dirichlet. The problem, however, is that as the Doppler bins (*N*) increased, the PAPR of the OTFS-BFDM system becomes unbearably high. For convenience, OTFS using rectangular PWFs can be represented by “OTFS-Rect”. Similarly, the terms “OTFS-RRC”, “OTFS-RC”, and “OTFS-Dirichlet” can also be obtained.

The PAPR of the OTFS-BFDM signal s(ℓ) on each frame is defined as:(15)$$PAPR=10\log\frac{\max_{0\leq \ell \leq O\cdot (L-1)}\;\left\{|s\left(\ell \right){|}^{2}\right\}}{E\{|s\left(\ell \right){|}^{2}\}},$$
where *O* is the oversampling factor and E{·} denotes the mathematical expectation. In this paper, *O* is assumed to be 4. The complementary cumulative distribution function (CCDF) is adopted to evaluate the PAPR performance of the OTFS-BFDM signal. The CCDF can be expressed by calculating the probability that the PAPR value of each sample exceeds the predefined threshold $PAPR_{0}$ as follows [24]:(16)$$C_{PAPR}\left(PAPR_{0}\right)=\mathrm{Pr}\left(PAPR>PAPR_{0}\right)$$

Figure 2a reveals the PAPR performance of OTFS-BFDM with different *N*, where M=32 and the symbol mapping is 4-QAM. For any value of *N*, when α=0 (α is the roll-off factor of RRC or RC), the CCDF curve of PAPR of OTFS-BFDM system with any PWFs is almost the same since the peak values of these PWFs in the time domain are close. When N=4, at a given CCDF $=10^{-3}$, the $PAPR_{0}$ of OTFS-Drichlet signal is 1 dB smaller than the OFDM signal. Furthermore, when N=16, the $PAPR_{0}$ of the OTFS-Dirichlet signal is almost the same as the OFDM signal. However, when N=64, the $PAPR_{0}$ of the OTFS-Dirichlet signal is 0.3 dB larger than the OFDM signal. This confirms that as *N* increases, the PAPR of the OTFS-BFDM signal becomes higher. When *N* increases to a certain extent, the high PAPR of the OTFS-BFDM signal will affect the system performance.

Figure 2b illustrates the PAPR performance of OTFS-BFDM with different PWFs, where M=32, N=31, and the symbol mapping is 4-QAM. At any given CCDF, the $PAPR_{0}$ of OTFS-BFDM with different PWFs is greater than OFDM. Moreover, the smaller the α is, the better PAPR performance of OTFS-BFDM signal is. This is because as α increases, the time-domain localization of the PWF becomes better and the peak value becomes larger.

Thus, it is necessary to take measures to suppress the PAPR of OTFS-BFDM signals. In this paper, the OCF method is preferred without compromising the sparsity of the ECM.

## 3. Proposed MP-AIC Scheme of Clipping Noise

### 3.1. Statistical Properties of Clipping Noise

Based on the central limit theory, s(ℓ) can be approximated as a complex Gaussian process with sufficiently large discrete samples, which has zero mean and variance $\sigma^{2}$. When a soft limiter is used to clip s(ℓ), the clipped signal can be written as:(17)$$s^{\prime }\left(\ell \right)=\left\{\begin{array}{c} Ae^{j\phi (t)},\left|s(\ell )\right|>A\\ \;s\left(\ell \right),\left|s(\ell )\right|\leq A \end{array}\right.$$
where ϕ(t) represents the phase of s(ℓ) and *A* is the clipping level.

As mentioned in [26], the signal after clipping is referred to as a clipped signal. According to the time-domain characteristics of the clipped signal, the clipped signal can be described as the sum of the original signal and the clipping distortion:(18)$$s^{\prime }\left(\ell \right)=\beta s\left(\ell \right)+\delta \left(\ell \right),0\leq \ell \leq OL-1,$$

In (Equation 18), δ(ℓ) is the clipping distortion uncorrelated with s(ℓ), and the clipping attenuation factor β∈(0,1) is defined as:(19)$$\beta =1-e^{-\gamma }+\frac{\sqrt{\pi \gamma }}{2}erfc\left(\sqrt{\gamma }\right),$$
where $\gamma =\frac{A^{2}}{\sigma^{2}}$ is the clipping ratio, representing the ratio of the clipping threshold power to the average power of the clipped signal. The variance of the clipping noise can be obtained as [28]:

The variance of clipping distortion can be divided into two cases: one is Nyquist rate sampling (oversampling factor *O* = 1, the filtering is not required) and the other is oversampling. For the Nyquist rate case, the variance of the clipping noise can be obtained as [26]:(20)$$E\left[{\left|\delta (l)\right|}^{2}\right]=\sigma_{\delta }^{2}=\left(1-e^{-r}-\beta^{2}\right)\sigma^{2}.$$

For the oversampling case, the clipping distortion of the out-of-band part is filtered out. The average power of distortion term can no longer be obtained by simply adding or subtracting based on in-band power, but by integrating the power spectral density (PSD) of the clipping distortion. The PSD of the clipping distortion can be obtained as [29]:(21)$$Sf_{\delta }\left(v\right)=FFT\left(R_{\delta }\left(u\right)\right),$$
where $R_{\delta }\left(u\right)=E\left[\delta_{\ell }\cdot \delta_{\ell +u}\right]=\sum_{n=1}^{OL}C_{\ell }{\left[\frac{R_{s}\left(u\right)}{R_{s}\left(0\right)}\right]}^{2\ell +1}$, $R_{\delta }\left(u\right)$ and $R_{s}\left(u\right)$ are autocorrelation functions of distortion term and original signal, respectively, and FFT represents the fast Fourier transform. The coefficient $C_{\ell }$ can be seen in [29], which depends on the clipping ratio γ only.

Then, the variance of the clipping distortion can be obtained by integrating $Sf_{\delta }\left(v\right)$ within the bandwidth, that is, take norm-1 for vector $Sf_{\delta }$ and then multiply it by Δf.

Referring to (Equation 11), the clipped OTFS-BFDM signal after downsampling can be given as:(22)$$\tilde{s}=H\left(\beta s+\delta \right)+w,$$
where $\delta ={\left[\delta (0),\delta (1),\cdots \;,\delta (L-1)\right]}^{T}$ and $w={\left[w(0),w(1),\cdots ,w(L-1)\right]}^{T}$.

At the receiver, the signal $\tilde{s}$ is first de-attenuated by multiplying $\beta^{-1}$. Then, the estimated signal $\hat{d}$ in the DD domain can be obtained through by (Equation 13), as follows:(23)$$\begin{array}{cc} \hat{d} & =\Lambda^{-1}\left(F_{N}⊗I_{M}\right)H\left(F_{N}^{-1}⊗I_{M}\right)\Lambda d\\ & +\underset{\Xi }{\underbrace{\beta^{-1}\Lambda^{-1}\left(F_{N}⊗I_{M}\right)H\delta }}+\underset{\;\;\;\psi \;\;\;}{\underbrace{\beta^{-1}\Lambda^{-1}\left(F_{N}⊗I_{M}\right)w}}, \end{array}$$
where Ξ is the clipping noise and ψ is the channel noise. Based on the linear properties of the mean and variance, $\;\;\;\Xi \;\;\;∼CN(0,\beta^{-2}\sigma_{n}^{2}I)$ can be obtained from [27]. Since each path on the time-varying multipath channel as described in (Equation 10) is a Rayleigh channel, the mean over all paths is 0 and the sum of the variances is 1. Combining (Equation 20), we get that $\;\;\;\psi \;\;\;∼CN(0,\beta^{-2}\sigma_{\delta }^{2}I)$.

Compared with (Equation 12) and (Equation 13), the ECM in (Equation 23) is unchanged and remains sparse. The overall noise variance in the clipped OTFS-BFDM system is:(24)$$\sigma_{All}^{2}=\beta^{-2}\left(\sigma_{\delta }^{2}+\sigma_{n}^{2}\right),$$
which serves as the noise input parameter of MP decoding algorithm.

### 3.2. Iterative Clipping Noise Cancellation Scheme

The main idea of the iterative clipping noise cancellation scheme is to cancel the clipping noise from the received signal by iteratively feeding the extracted residual signal back to the MP detector. As the number of iterations increases, the accuracy of MP decoding will be improved and the residual amount of clipping noise in the initial received signal is finally canceled.

Figure 3 describes the iterative clipping noise cancellation with a maximum iteration number of $\tau_{\max}$ in the clipped OTFS-BFDM system. The scheme includes the following steps:(1)The received signal $\tilde{s}$ is first de-attenuated by multiplying $\beta^{-1}$. Then, BFDM demodulation and SFFT transformation are performed on the attenuated signal to obtain the observation signal $\hat{d}$. Finally, the decision signal $\tilde{d}$ will be output when $\hat{d}$ is substituted into MP algorithm for equalization and decoding.According to (Equation 13) and (Equation 22), $\hat{d}$ can be derived as:
(25)$$\begin{array}{cc} \hat{d} & =\left(F_{N}⊗I_{M}\right)\mathrm{QHP}\left(F_{N}^{-1}⊗I_{M}\right)d+\beta^{-1}\left(F_{N}⊗I_{M}\right)QH\delta +\beta^{-1}\left(F_{N}⊗I_{M}\right)\mathrm{Qw}. \end{array}$$(2)Let the decision output $\tilde{d}$ passes through the BFDM modulator and ISFFT converter in turn to obtain a new OTFS-BFDM signal. The new OTFS-BFDM signal will be processed in two parallel ways. One way is to apply the same clipping and filtering as the transmitter to obtain a new clipped signal ${\tilde{s}}_{1}$. The other way is to multiply it by β to get the attenuated signal ${\tilde{s}}_{2}$. The constructed reference clipping distortion $\delta^{\prime }$ can be generated by subtracting ${\tilde{s}}_{2}$ from ${\tilde{s}}_{1}$.According to (Equation 7) and (Equation 18), we get:
(26)$${\tilde{s}}_{1}=\beta P\left(F_{N}^{-1}⊗I_{M}\right)\tilde{d}+\delta^{\prime },{\tilde{s}}_{2}=\beta P\left(F_{N}^{-1}⊗I_{M}\right)\tilde{d}.$$It is evident that $\delta^{\prime }={\tilde{s}}_{1}-{\tilde{s}}_{2}$. Note that the constructed reference clipping distortion $\delta^{\prime }$ has to go through the channel convolution matrix H before the next iteration, where H is known due to the ideal channel estimation.(3)Remove the clipping distortion $H\delta^{\prime }$ from the received signal $\tilde{s}$ to obtain the revised received signal $\bar{s}$.From Figure 3, the signal $\bar{s}$ can be written as:
(27)$$\bar{s}=\beta \mathrm{HP}\left(F_{N}^{-1}⊗I_{M}\right)d+H\left(\delta -\delta^{\prime }\right)+w,$$
where $\delta -\delta^{\prime }$ represents the difference in clipping distortion.(4)Replace $\tilde{s}$ with $\bar{s}$ and return to step 1 for the next iteration. Until the number of iterations reaches $\tau_{\max}$, the procedure terminates.In the first iteration, by attenuating and demodulating the signal $\bar{s}$ in turn, the revised observation signal $\bar{d}$ can be obtained as:
(28)$$\begin{array}{cc} \bar{d} & =\left(F_{N}⊗I_{M}\right)QHP\left(F_{N}^{-1}⊗I_{M}\right)d+\underset{Residual\_signal}{\underbrace{\beta^{-1}\left(F_{N}⊗I_{M}\right)QH\left(\delta -\delta^{\prime }\right)}}+\beta^{-1}\left(F_{N}⊗I_{M}\right)Qw. \end{array}$$In (Equation 28), $\beta^{-1}\left(F_{N}⊗I_{M}\right)QH\left(\delta -\delta^{\prime }\right)$ is the residual signal and will be further reduced in the next iteration.

### 3.3. The Procedure of the MP-AIC Method

In the original method of iterative clipping noise cancellation, the residual clipping noise is gradually canceled and the BER tends to converge. However, the initial probability of modulation alphabet used in MP decoding is still equal after each clipping noise cancellation. In the MP-AIC method, we use the variance of the residual signal and channel noise together as input to the MP algorithm and use the probability of modulation alphabet after MP decoding in the current iteration as the initial probability of MP decoding in the next iteration to accelerate the convergence rate of MP decoding. Therefore, the proposed MP-AIC method can further improve the BER of the clipped system and accelerate the BER convergence.

Assume that $H_{eq}$ has only *S* non-zero elements in each row and column. Each observation node ${\tilde{d}}_{a}$ is connected to the set of variable nodes $\left\{d_{a},a\in S\right\}$, and each variable node $d_{b}$ is connected to the set of observation nodes $\left\{{\tilde{d}}_{b},b\in S\right\}$. The maximum a posteriori (MAP) decision rule for (Equation 13) is given by:(29)$$\hat{d}=\underset{d\in A^{L\times 1}}{\arg\max}\;\mathrm{Pr}\left(d|\tilde{d},H_{eq}\right),$$
where A is a modulation alphabet of size *Q*. Using the Bayesian criterion, the MAP joint detection output is:(30)$${\hat{d}}_{a}=\underset{a_{j}\in A}{\arg\max}\;\prod_{b\in S}\;\mathrm{Pr}\left({\tilde{d}}_{b}|d_{a}=a_{j},H_{eq}\right).$$

In the MP algorithm, the message passed from a variable node $d_{a}$ to the observation nodes ${\tilde{d}}_{b}$ is the probability mass function (pmf) of the alphabet $P_{ab}=\left\{p_{ab}\left(a_{j}\right)|a_{i}\in A\right\}$.

The procedure of the MP-AIC method is expressed as follows:(1)Input: clipping attenuation factor β, Doppler tap $k_{p}$, delay tap $l_{p}$, path gain $h_{p}$, iteration number $N_{iter}$ (Iterative Clipping Noise Cancellation ), iteration number $\tau_{\max}$ (Iterative MP decoding), Gaussian white noise variance $\sigma_{n}^{2}$, the damping factor Δ=0.6, ε=0.01.(2)Initialization: overall noise variance $\sigma {{}_{All}^{2}}_{\left(0\right)}=\sigma {}_{All}^{2}$ (Equation (Equation 24)), the initial probability of constellation symbols $p_{ab}^{\left(0\right)}=1/Q$ and the observation siganl ${\hat{d}}_{\left(0\right)}=\hat{d}$ (Equation (Equation 25)) in the zero-th iteration.(3)Calculate the total noise variance and observation signal of the *k*-th iteration by $\sigma {{}_{All}^{2}}_{\left(k\right)}=\beta^{-2}{(\sigma_{\delta }^{2}}_{\left(k\right)}+\sigma_{n}^{2})$ (Equations (Equation 20), (Equation 24), and (Equation 28)). ${\hat{d}}_{\left(k\right)}=\beta^{-1}{\tilde{s}}_{\left(k\right)}$ (Equations (Equation 25) and (Equation 27)).(4)Message from ${\tilde{d}}_{b}$ to $x_{a}$: The mean $\mu_{ba}^{\left(i\right)}$ and variance ${\left(\sigma_{ba}^{\left(i\right)}\right)}^{2}$ of the interference term $I_{ba}$ are passed as messages from ${\tilde{d}}_{b}$ to $x_{a}$:
(31)$$I_{ba}=\sum_{c\in S,c\neq a}d_{c}H_{b,c}+\xi_{b}.$$The mean and variance of $I_{ba}$ are expressed, respectively, as:
(32)$$\mu_{ba}^{\left(i\right)}=\sum_{c\in S,c\neq a}\sum_{j=1}^{Q}{p_{cb}^{\left(i\right)}}_{\left(k\right)}\left(a_{j}\right)a_{j}H_{b,c},$$
and:
(33)$${\left(\sigma_{ba}^{\left(i\right)}\right)}^{2}=\sum_{\begin{array}{c} c\in S\\ c\neq a \end{array}}\sum_{j=1}^{Q}{p_{cb}^{\left(i\right)}}_{\left(k\right)}\left(a_{j}\right)\left|a_{j}{|}^{2}\right|H_{b,c}{|}^{2}-\sum_{\begin{array}{c} c\in S\\ c\neq a \end{array}}{\left|\sum_{j=1}^{Q}{p_{cb}^{\left(i\right)}}_{\left(k\right)}\left(a_{j}\right)a_{j}H_{b,c}\right|}^{2}+{\sigma_{all}^{2}}_{\left(k\right)},$$
where *k* and *i* represent iterations of IC and MP, respectively.(5)Message from $x_{a}$ to ${\tilde{d}}_{b}$: The pmf vector ${p_{ab}^{\left(i+1\right)}}_{\left(k\right)}$ can be updated as:
(34)$${p_{ab}^{\left(i+1\right)}}_{\left(k\right)}=\Delta {p_{ab}^{\left(i\right)}}_{\left(k\right)}\left(a_{j}\right)+\left(1-\Delta \right)\Delta {p_{ab}^{\left(i-1\right)}}_{\left(k\right)}\left(a_{j}\right),$$
where:
(35)$${p_{ab}^{\left(i\right)}}_{\left(k\right)}\propto \prod_{c\in S,c\neq b}\mathrm{Pr}({\tilde{d}}_{c}|d_{a}=a_{j},H_{eq}),$$
and:
(36)$$\begin{array}{cc} \; & \mathrm{Pr}\left(\hat{d}|d_{a}=a_{j},H_{eq}\right)\propto \exp\left(\frac{-{\left|{\hat{d}}_{c}-\mu_{ca}^{\left(i\right)}-H_{c,a}a_{j}\right|}^{2}}{\sigma_{c,a}^{2(i)}}\right). \end{array}$$(6)Return to step 4 until $\max_{a,b,a_{j}}\;\left|p_{ab}^{\left(i+1\right)}\left(a_{j}\right)-p_{ab}^{\left(i\right)}\left(a_{j}\right)\right|\leq \varepsilon$ or $i\geq \tau_{\max}-1$.(7)Return to step 3 until $k\geq N_{iter}-1$.(8)Output the mp-decoded symbol as:
(37)$${\hat{d}}_{a}=\underset{a_{j}\in A}{\arg\max}\;p_{a}\left(a_{j}\right),a\in \left[0,L-1\right].$$

In conclusion, the pseudo-code of MP-AIC method for clipping noise cancellation can be expressed by Algorithm 1.
**Algorithm 1:** The procedure of the MP-AIC method.**Require:**β, $k_{p}$, $l_{p}$, $h_{p}$, $N_{iter}$, $\tau_{\max}$, $\sigma_{n}^{2}$, Δ=0.6, ε=0.01.**Ensure:**${\hat{d}}_{a}=\underset{a_{j}\in A}{\arg\max}\;p_{a}\left(a_{j}\right)$ and $p_{ab}^{\left(\tau_{\max}-1\right)}$.    **Initialize** k=0, j=0, $\sigma {{}_{All}^{2}}_{\left(0\right)}=\sigma {}_{All}^{2}$, $p_{ab}^{\left(0\right)}=1/Q$, ${\hat{d}}_{\left(0\right)}=\hat{d}$, ${\sigma_{\delta }^{2}}_{\left(0\right)}=\sigma_{\delta }^{2}$.    **while**
$\;k\leq N_{iter}-1$
**do**         $\sigma {{}_{All}^{2}}_{\left(k\right)}=\beta^{-2}{(\sigma_{\delta }^{2}}_{\left(k\right)}+\sigma_{n}^{2})$ (Equations (Equation 20), (Equation 24) and (Equation 28)).         ${\hat{d}}_{\left(k\right)}=\beta^{-1}{\tilde{s}}_{\left(k\right)}$ (Equations (Equation 25) and (Equation 27)).         **while** $\max_{a,b,a_{j}}\;\left|p_{ab}^{\left(i+1\right)}\left(a_{j}\right)-p_{ab}^{\left(i\right)}\left(a_{j}\right)\right|>\varepsilon$ or $i\leq \tau_{\max}-1$ **do**             **for** a=1 to *S* **do**                 **for** b=1 to *S* **do**                     Message from ${\tilde{d}}_{b}$ to $x_{a}$:                     $\mu_{ba}^{\left(i\right)}=\sum_{c\in S,c\neq a}\sum_{j=1}^{Q}{p_{cb}^{\left(i\right)}}_{\left(k\right)}\left(a_{j}\right)a_{j}H_{b,c}$                     ${\left(\sigma_{ba}^{\left(i\right)}\right)}^{2}=\sum_{c\in S,c\neq a}\sum_{j=1}^{Q}p{{}_{cb}^{\left(i\right)}}_{\left(k\right)}{\left|a_{j}\right|}^{2}{\left|H_{b,a}\right|}^{2}-\sum_{c\in S,c\neq a}{\left|\sum_{j=1}^{Q}p{{}_{cb}^{\left(i\right)}}_{\left(k\right)}\left(a_{j}\right)a_{j}H_{b,c}\right|}^{2}+\sigma {{}_{all}^{2}}_{\left(k\right)}$                 **end for**             **for** **end for**             **for** **for** a=1 to *S* **do**                 **for** b=1 to *S* **do**                     Message from $x_{a}$ to ${\tilde{d}}_{b}$:                     ${p_{ab}^{\left(i\right)}}_{\left(k\right)}\propto \prod_{c\in S,c\neq b}\mathrm{Pr}({\tilde{d}}_{c}|d_{a}=a_{j},H_{eq})$                     ${p_{ab}^{\left(i+1\right)}}_{\left(k\right)}=\Delta {p_{ab}^{\left(i\right)}}_{\left(k\right)}\left(a_{j}\right)+\left(1-\Delta \right)\Delta {p_{ab}^{\left(i-1\right)}}_{\left(k\right)}\left(a_{j}\right)$                 **end for**              **end for**              i=i+1           **end while**           k=k+1         **end while**

## 4. Computational Complexity

In [26], the MP-A method only needs one *N*-IFFT/FFT and two *M*-IFFT/FFT transforms, as well as one MP operation. However, in the proposed MP-AIC scheme, each iteration requires the same complexity as the MP-A method. Beyond that, with the increase in iterations, the computational complexity of the MP-AIC method will increase substantially. Fortunately, the simulation results in the next section will show that two iterations can ensure the BER close to convergence, which is beneficial to the application of the MP-AIC method.

The complexity of MP-AIC mainly exists in the message passing between observation nodes and variable nodes [7]. Moreover, an *N*-point IFFT requires $N\log_{2}N$ multiplications and $\frac{1}{2}N\log_{2}N$ additions. Assume that $H_{eq}$ has only *S* non-zero elements in each row and column, the size of modulation alphabet is *Q* and the number of iterations of MP algorithm is $\tau_{\max}$. To update variable nodes from observation nodes, $LSQ^{S}\left(2S+3\right)$ multiplications and $LS^{2}\left(Q^{S}-Q\right)$ additions are involved, while to update variable nodes from observation nodes, only LSQ(S−2) multiplications are need. Since the maximum number of iterations of the MP-AIC method is $N_{iter}$, the total computational complexity can be calculated as:(38)$$\begin{array}{cc} \; & N_{mul}=\tau_{\max}N\log_{2}N+2\tau_{\max}M\log_{2}M+\tau_{\max}\left(N_{iter}+1\right)\left(LSQ^{S}\left(2S+3\right)+LSQ\left(S-2\right)\right), \end{array}$$
(39)$$\begin{array}{cc} \; & N_{add}=\frac{\tau_{\max}}{2}N\log_{2}N+\tau_{\max}M\log_{2}M+\tau_{\max}\left(N_{iter}+1\right)\left(LS^{2}\left(Q^{S}-Q\right)\right), \end{array}$$
where $N_{mul}$ and $N_{add}$ are the total complexity of multiplication and addition, respectively.

## 5. Simulation Results

In this section, we evaluate the performance of the clipped OTFS-BFDM system by simulations. Firstly, the PAPR distribution for clipped OTFS-BFDM signals is given, then the BER performance between the MP-AIC and MP-A methods in the clipped system is compared. In the simulation below, the delay-Doppler profile considered is shown in Table 2, while other relevant simulation parameters are summarized in Table 3, where the channel has five taps with uniform power [30].

### 5.1. CCDFs of the PAPR for OTFS-BFDM Signals with Different γ

Figure 4 shows the CCDFs of the PAPR for OTFS-BFDM signals with different clipping ratios γ. As shown in Figure 4a, the PAPR of OTFS-Rect signal is unbearable high. After clipping, the PAPR can be significantly reduced. The $PAPR_{0}$ of non-clipped OTFS-Rect signals is 11.5 dB at $CCDF=10^{-3}$. After clipping with γ=3 dB, the $PAPR_{0}$ can be reduced by 4.7 dB. Moreover, the smaller the clipping ratio, the more significant the PAPR decreases since the clipping ratio is proportional to the clipping threshold. A similar conclusion can be drawn from Figure 4b.

### 5.2. BER Comparison between the MP-AIC and MP-A Schemes

Figure 5 shows the BER of the non-clipped OTFS-BFDM system with different PWFs. In Section 2.2, we learned that the better the frequency-domain localization of non-rectangular PWFs, the lower the PAPR of the OTFS-BFDM system. When the OTFS-BFDM system adopts RRC/RC PWFs with α=0 or Dirichlet PWFs, their PAPR performance is consistent with that of the OTFS-Rect system. To verify that the BER performance of OTFS-BFDM system using non-rectangular PWFs is not inferior to that of OTFS-Rect system in high mobility scenarios, the relationship between PWFs and BER of the system can also be obtained through simulation. In Figure 5a, we adopt the MP [7] and linear minimum mean square error (LMMSE) [1] equalization to simulate the OTFS-BFDM system with different PWFs in three scenarios. It can be seen that when the OTFS-BFDM system adopts rectangular PWFs, RRC/RC PWFs (α=0) and Dirichlet PWFs, respectively, the BERs of the system are almost the same regardless of the symbol mapping, channel equalization and maximum Doppler shift, which is consistent with the conclusion in [14]. It is worth noting that the BER of the linear equalizer LMMSE is higher than that of MP equalizer because it cannot obtain the time-frequency diversity of the channel. When the symbol modulation is 4-QAM and the maximum Doppler shift is 967.74 Hz, the BER of LMMSE equalizer and MP equalizer are $1.563\times 10^{-4}$ and $2.974\times 10^{-4}$, respectively, at SNR=15 dB. Figure 5b depicts the BER performance of an OTFS-RRC system at different roll-off coefficients α, where the smaller the α, the smaller the BER of the system. When the symbol modulation is 4-QAM and the maximum Doppler shift is 967.74 Hz, the BERs of OTFS-RRC system with α=0.1 and α=0.9 are $2.6\times 10^{-4}$ and $7.681\times 10^{-3}$, respectively, at SNR=16 dB. Therefore, the better the frequency-domain localization of non-rectangular PWFs, the better the BER of OTFS-BFDM system. In particular, the BER of OTFS-BFDM system using (α=0) and Dirichlet PWFs is almost the same as that of OTFS-Rect system.

Figure 6 shows the BER performance of the MP-A method for clipping noise suppression in [26]. Although the MP-A method considers the influence of clipping noise on MP algorithm, it only alleviates the clipping noise interference but does not cancel it, where MP-A takes clipping noise and channel noise as the whole noise input of MP algorithm. For the case of Figure 6a, the BER of OTFS-Rect system deteriorates obviously after clipping and increases with the clipping ratio γ. When the MP-A method is used to suppress the clipping noise, the BER peformance is slightly improved. On the channel with the maximum Doppler shift of 1935.48 Hz, when γ=1 dB and SNR=16 dB, the BER is $3.717\times 10^{-3}$, while the BER in MP-A method is $2.701\times 10^{-3}$. It can be concluded from the above that MP-A can alleviate the interference of clipping noise to MP decoding to a certain extent, but cannot completely cancel the interference. A similar conclusion can be drawn from Figure 6b.

As shown in Figure 7a, the BER of the MP-AIC method decreases with the increase of iterations, because the residual clipping noise is gradually canceled out after several iterations. When the number of iterations increases from 0 to 2, the BERs of MP-AIC are $5.814\times 10^{-3}$, $1.809\times 10^{-3}$, and $1.62\times 10^{-3}$, respectively, at γ=1 dB and SNR = 16 dB. It can be seen that the BER at iter=2 is about 72% lower than that at iter=0 since clipping noise is not considered at iter=0. In addition, with the increase of γ, the BER of MP-AIC also decreases, because the lager γ is, the less distortion of the signal. When γ=1and3 dB, the BER of the MP-AIC with two iterations are $1.62\times 10^{-3}$ and $9.194\times 10^{-4}$, respectively, at SNR=16 dB, while those of MP-A are, respectively, $3.472\times 10^{-3}$ and $2.701\times 10^{-3}$. Then, the BER of MP-AIC is 53% lower than that of MP-A at γ=1 dB. Similarly, in Figure 7b, when γ=1and3 dB, the BER of MP-AIC with two iterations are $2.828\times 10^{-3}$ and $1.817\times 10^{-3}$, respectively, at SNR of 16 dB, while those of MP-A are, respectively, $8.007\times 10^{-3}$ and $3.476\times 10^{-3}$. Then, the BER of MP-AIC is about 47.7% lower than that of MP-A at γ=1 dB. To sum up, the BER performance of MP-AIC is much better than that of MP-A in canceling clipping noise interference.

Figure 8 shows the BER convergence diagram of the clipped OTFS-RRC system under different iterations of the MP-AIC method, where α=0.1. It can be seen that with the increase of SNR, the MP-AIC method improves the BER performance more and more significantly. Furthermore, in the first iteration, the BER decreases the most, while after two iterations, the BER tends to converge. Referring to (Equation 25), most of the clipping noise is reconstructed and canceled in the first iteration and only the remaining residual clipping noise terms are gradually reduced in the second and more iterations. Therefore, two iterations can ensure that the BER performance of system is close to optimal. Similarly, for clipped OTFS-BFDM signals with different PWFs, the MP-AIC method also has a consistent convergence effect on BER.

## 6. Conclusions

In this paper, we propose an MP-AIC method to cancel the clipping noise at the receiver. This method is combined with the OCF at the transmitter to maintain the sparsity of ECM. Therefore, the low complexity MP algorithm can take advantage of the sparsity of ECM to achieve joint interference cancellation and signal decoding. To minimize the influence of clipping noise on MP decoding accuracy, the reference clipping noise is constructed on the receiver. The residual amount of clipping noise in the initial received signal is canceled by iteration. The simulation results show that the MP-AIC method provides better BER performance than MP-A, and the BER curve of the system is close to convergence after two iterations.

With the increasing pursuit of spectral efficiency in 5G and above communication systems, high-order modulation will also be widely used. However, high-order modulation makes the transmission signal more sensitive to clipping distortion. Therefore, in future work, we will introduce non-distortion techniques, such as signal scrambling or coding techniques, to suppress the PAPR of OTFS-BFDM signals. Moreover, we foresee that how to achieve an optimal performance tradeoff between PAPR and BER in high mobility scenarios will be one of the hot topics of future research.

## Figures and Tables

**Figure 1 sensors-22-03937-f001:**
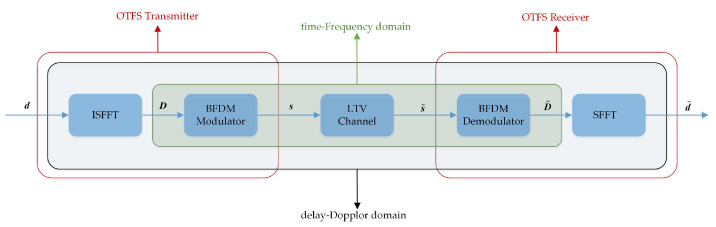
OTFS-BFDM system model.

**Figure 2 sensors-22-03937-f002:**
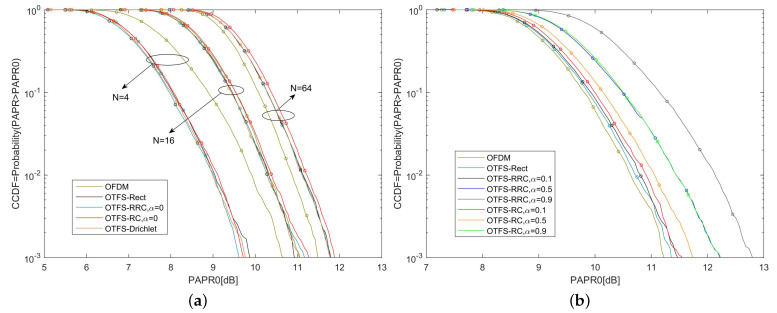
(**a**) The CCDF of PAPR of OTFS-BFDM systems with different Doppler bins (*N*); (**b**) the CCDF of PAPR of OTFS-BFDM systems with different PWFs.

**Figure 3 sensors-22-03937-f003:**
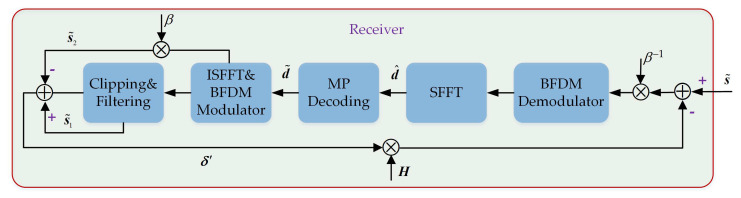
The iterative cancellation of clipping noise for OTFS-BFDM signals on the receiver.

**Figure 4 sensors-22-03937-f004:**
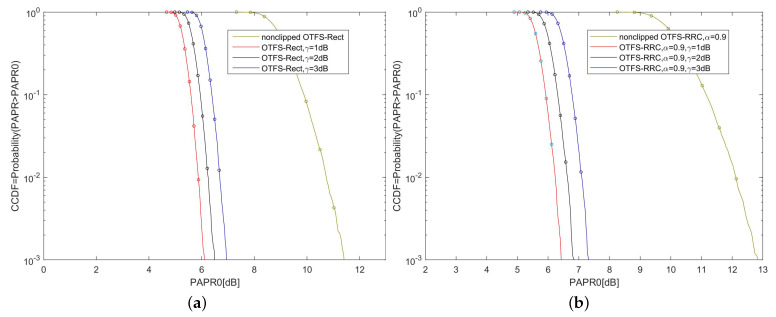
(**a**) PAPR distribution for the OTFS-Rect signal with different clipping ratio γ; (**b**) PAPR distribution for the OTFS-RRC signal with different clipping ratio γ (α=0.9).

**Figure 5 sensors-22-03937-f005:**
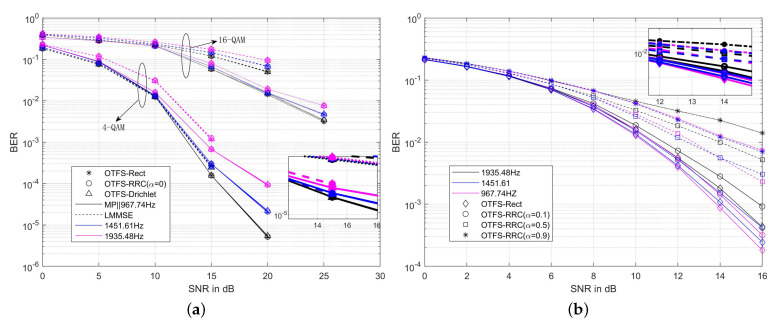
(**a**) BER comparison of the non-clipped OTFS-BFDM system with rectangular, RRC (α=0), and Drichlet PWFs; (**b**) BER comparison of the non-clipped OTFS-RRC system with α=0.1, α=0.5, and α=0.9. The channels in both scenarios contain the maximum Doppler shifts of 979.74 Hz, 1451 Hz, and 1935.48 Hz.

**Figure 6 sensors-22-03937-f006:**
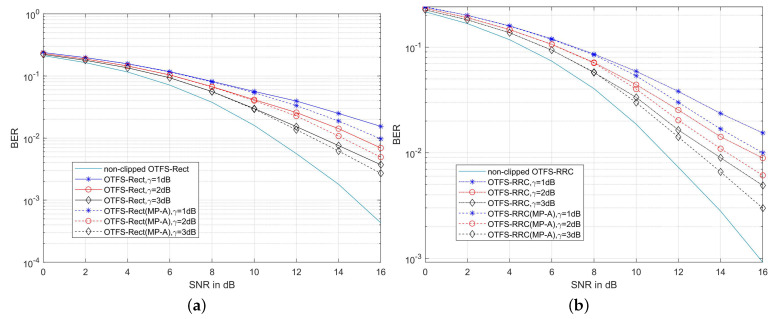
(**a**) BER performance of the MP-A [26] method in the clipped OTFS-Rect system on the channel with a maximum Doppler shift of 1935.48 Hz; (**b**) BER performance of the MP-A [26] method in the clipped OTFS-RRC system on the channel with a maximum Doppler shift of 1935.48 Hz ((α=0.1)).

**Figure 7 sensors-22-03937-f007:**
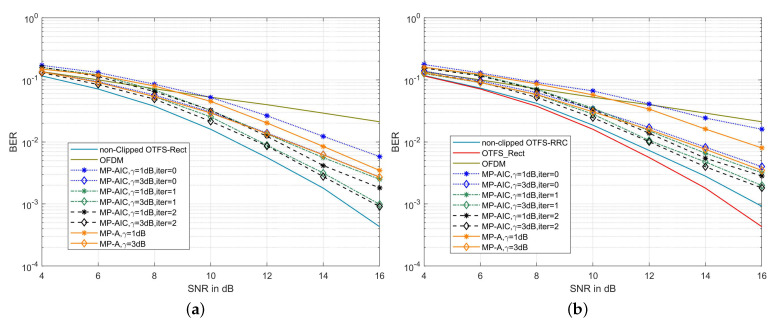
(**a**) BER comparison of the proposed MP-AIC method and MP-A [26] method in the clipped OTFS-Rect system on the channel with a maximum Doppler shift of 1935.48 Hz; (**b**) BER comparison of the proposed MP-AIC method and MP-A [26] method in the clipped OTFS-RRC system on the channel with a maximum Doppler shift of 1935.48 Hz (α=0.1).

**Figure 8 sensors-22-03937-f008:**
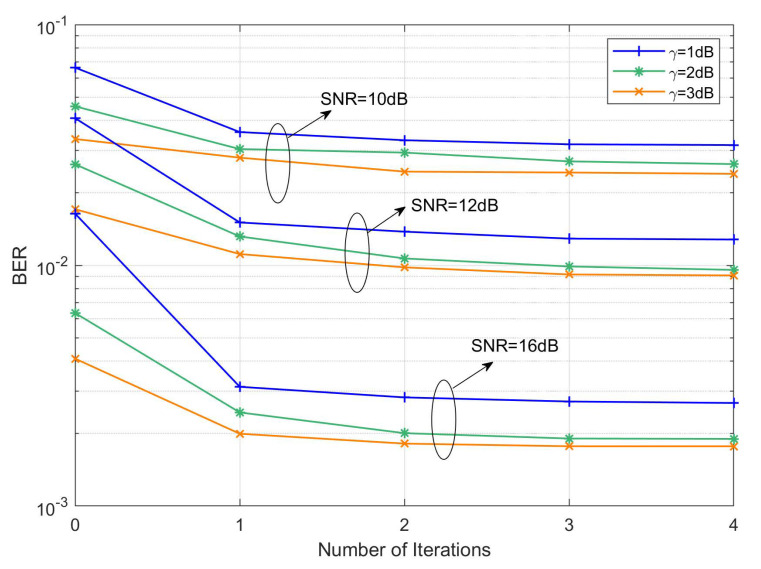
BER convergence of a clipped OTFS-BFDM system with RRC PWFs under different iterations (α=0.1).

**Table 1 sensors-22-03937-t001:** Mathematical notation.

Notation	Description
${\left(*\right)}^{-1}/{\left(*\right)}^{T}$	The inverse/transpose of a matrix
⊗	The kronecker product
$I_{M}$	A *M*-dimensional identity matrix
$F_{N}$	A *N*-point DFT matrix ${\left\{\frac{1}{\sqrt{N}}e^{2\pi jkl/N}\right\}}_{k,l=0}^{N-1}$
$F_{N}^{-1}$	A *N*-point IDFT matrix ${\left\{\frac{1}{\sqrt{N}}e^{-2\pi jkl/N}\right\}}_{k,l=0}^{N-1}$
diag{*}	Return a diagonal matrix
circ{*}	Return a circular matrix

**Table 2 sensors-22-03937-t002:** Delay-Doppler profile for the channel model with path = 5.

Path Index	1	2	3	4	5
Delay (μs)	0	2.08	4.17	6.25	8.33
Doppler_1 (Hz)	0	0	0	483.87	967.74
Doppler_2 (Hz)	0	0	483.87	967.74	1451.61
Doppler_3 (Hz)	0	483.87	967.74	1451.61	1935.48

**Table 3 sensors-22-03937-t003:** Simulation parameters.

Parameters	Value
Carrier frequency (GHz)	4
Sub-carrier spacing (kHz)	15
Frame size (M,N)	(32, 31)
Modulation scheme	4-QAM
CP (μs)	10.42
Channel estimation	ideal
Oversampling factor	4

## Data Availability

Not Applicable, the study does not report any data.

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
