# Peer review of "A Message Passing-Assisted Iterative Noise Cancellation Method for Clipped OTFS-BFDM Systems"

_sensors, 2022, doi:10.3390/s22103937_

Round 1
Reviewer 1 Report
This paper adds valuable information and appropriately structured the research. However, the authors have to address all of the below concerns carefully.
- Paper title: It contains some acronyms (such as MP and OTFS-BFDM) that makes it unclear. We prefer to replace them with words.
- Abstract: Numerical results should be added at the end of the abstract showing the importance of the proposed approach.
- The computational Complexity Section contained comparisons of some approaches but does not include comparisons between the results of the proposed research and the results of the existing research included in this paper.
- Conclusions Section: Future directions should be added.
- Figures: Some figures require enlarging such as Figures 2, 4, 5, 6 ... etc. Some figures are shown before they are used in the text.
- References list: References should follow the MDPI-Sensors style. For instance, some search names in the reference list begin an uppercase letter for each word (such as [1], [3], [4] ... etc.) and others use only an uppercase letter in the first word (such as [2], [12], [14] … etc.), author should standardize style. The number of references is insufficient for this research. Some references do not contain enough information such as reference [8] … etc. The references list requires moderate scrutiny by the authors.
- English Writing: This paper requires moderate proofreading. There are some of grammatical, spelling and typos problems. The authors have to thoroughly scrutinize the paper. Without professional, accurate and clear English, readers cannot understand the research.
Reviewer 2 Report
In this manuscript, the authors use a simplified clipping and filtering (SCF) approach to reduce the peak-to-average ratio (PAPR) of OTFS-BFDM signals. They propose and describe in detail a message passing (MP)-assisted iterative cancellation algorithm to minimize the clipping noise. Moreover, they verify the developed method through simulation results and comparison with existing baselines in terms of BER and they concretely show that two iterations of their method can ensure a close-to-optimal BER performance. The manuscript is in general well-written, the presented work is interesting and the references are appropriate, permitting a comprehensive follow-up of the problem. Some minor comments:
- There are several typos, e.g. result instead of "rusult", page 1 line 24, have instead of “has” in line 27, Dirichlet instead of “Drichlet”. Please carefully proofread your manuscript.
- Please enlarge the fonts and the labels in all Figures (especially Figures 4 – 8) to enhance their quality and improve their readability.
Reviewer 3 Report
The authors of this article proposed an iterative cancelation method, called message passing assisted iterative cancelation (MP-AIC), at a receiver to reduce the signal distortion introduced from clipping and filtering to reduce the PAPR at a transmitter for OTSF-BFDM systems. The manuscript is organized very well and written clearly. The representation of the OTSF-BFDM system and derivation of the noise term due to clipping and filtering are well formulated. The high PAPR is one of known problems for OTSF based system. The technique proposed by the authors provides an alternative solution to solve this problem. To make the proposed technique be useful in future 6G systems, the authors should consider more practical situations than the assumptions made in this article.
Additional comments below:
- One of the shortcomings of Clipping and Filtering method is high OOBE, which is also shown in Figure 5 of this article. It would be better for authors to discuss this issue and, even better, to use some mask requirements in industry standards (e.g., 3GPP or Wi-Fi) to see if the observed OOBE can meet those requirements.
- One important assumption in the representation of the system model and simulation is that the cannel matrix is ideal. It would better for the authors to discuss the impact of non-ideal channel information to the proposed technique.
- When comparing different methods using simulation in terms the effect of PAPR reduction to BER, nonlinear PA with proper power back off should be included.
- In section 3.1, where the SCF is introduced, it is not clear if Equation (18) can used for all possible s(l) values or just for |s(l)| > A. If it is former, the \beta and \sigma(l) should also be defined based on the cases.
- “Convergence probability” is a confusing term. It sounds like the probability of convergence. But it doesn’t seem the case. The authors may want to explain it and also discussion how to ensure the convergence of the cancelation.
- When using comparative adjectives, e.g., larger or longer, one should always mention what is compared. For example, in Page 5, Line 117, the authors wrote “The OTFS-BFDM system extends the waveform flexibility by using longer non-rectangular PWFs such as root raised cosine (RRC), raised cosine (RC) and Drichlet.” Longer than what should be mentioned. Authors should check other places in this article with the same issue.
